# Comparative Cutaneous Water Loss and Desiccation Tolerance of Four *Solenopsis* spp. (Hymenoptera: Formicidae) in the Southeastern United States

**DOI:** 10.3390/insects11070418

**Published:** 2020-07-05

**Authors:** Olufemi S. Ajayi, Arthur G. Appel, Li Chen, Henry Y. Fadamiro

**Affiliations:** 1Department of Entomology and Plant Pathology, Auburn University, Auburn, AL 36849, USA; fadamhy@auburn.edu; 2Department of Bioinformatics, College of Life Science, Institute of Life Science and Green Development, Hebei University, Baoding 071002, China; chenli1@hbu.edu.cn

**Keywords:** cuticular permeability, desiccation resistance, *Solenopsis invicta*, *Solenopsis richteri*, *Solenopsis invicta* × *S*. *richteri*, *Solenopsis geminata*, distribution

## Abstract

The high surface area to volume ratio of terrestrial insects makes them highly susceptible to desiccation mainly through the cuticle. Cuticular permeability (CP) is usually the most important factor limiting water loss in terrestrial insects. Water loss rate, percentage of total body water (%TBW) content, CP, and desiccation tolerance were investigated in workers of four *Solenopsis* species in the southeastern USA. We hypothesized that tropical/subtropical ants (*S*. *invicta* and *S*. *geminata*) will have lower CP values and tolerate higher levels of desiccation than temperate ants (*S*. *richteri* and *S. invicta* × *S*. *richteri*). The %TBW content was similar among species. *Solenopsis invicta* had a 1.3-fold and 1.1-fold lower CP value than *S. invicta* × *S. richteri* and *S. richteri*, respectively. *Solenopsis geminata* had a 1.3-fold lower CP value than *S*. *invicta* × *S*. *richteri*, and a 1.2-fold lower CP value than *S*. *richteri*. The LT_50_ values (lethal time to kill 50% of the population) ranged from 1.5 h (small *S. geminata*) to 8.5 h (large *S. invicta*). Desiccation tolerance ranged between 36 and 50 %TBW lost at death and was not related to a species’ location of origin. This study is the first report of water relations of *S. invicta* × *S. richteri*. It demonstrates that desiccation stress differentially can affect the survival of different *Solenopsis* species and implies that environmental stress can affect the distribution of these species in the southeastern USA.

## 1. Introduction

The distribution of terrestrial insects is limited by tolerance of abiotic stressors such as temperature and relative humidity [1,2,3]. Desiccation impacts insect water balance, and tolerance to this stress plays an important role in the geographic distribution of insects [4,5,6]. Understanding the native distribution of invasive insect species is useful information for predicting their potential spread in newly introduced areas [7,8]. Desiccation tolerance (i.e., percentage of total body water lost at death) can be expressed phenotypically between generations [9]. Desiccation tolerance in terrestrial insects is limited by apparent morphological and ecological factors [9,10,11].

The high surface area to volume ratio of terrestrial insects makes them highly susceptible to desiccation mainly through the cuticle [12]. Terrestrial insects conserve water by a relatively water impermeable cuticle covered with a thin layer of epicuticular lipids (particularly hydrocarbons) which serve as the primary mechanism to limit loss of water across the insect’s exoskeleton [5,13,14]. Cuticular permeability is usually the most important factor limiting water loss [13,14,15,16] and is often measured as the amount of water lost (µg) per unit surface area (cm^2^) per unit time (h) per unit saturation deficit (mmHg, Torr, or kPa) [13,15,16,17]. Higher cuticular permeability values represent greater rates of water loss and faster desiccation. Cuticular permeability of terrestrial insects is influenced by climatic factors such as temperature, relative humidity, and saturation deficit in their habitat [12,16]. These climatic factors vary between latitudes [17,18,19,20,21] and are being negatively impacted by climate change [22,23]. Dramatic variation in water availability between habitats is another important factor of species distribution. Cuticular permeability is also significantly affected by the condition (live or dead) of terrestrial insects because the cuticular water pump, located in the epidermal cells, actively restricts water loss in living insects [12,24,25,26]. These factors and variabilities are expected to play a role in limiting species’ geographical distribution. However, little is known about how desiccation tolerance is related to the distribution and success of invasive insect species in their introduced environment as compared with their native range [27].

The black imported fire ant, *Solenopsis richteri* Forel, and the red imported fire ant, *S*. *invicta* Buren, were accidentally introduced into the United States of America from South America in the 1920s and 1930s, respectively, through Mobile, Alabama [28,29]. Since their introduction, both *S. richteri* and *S*. *invicta* have become serious pests and *S. invicta* has displaced native fire ants [30], including the tropical fire ant, *S*. *geminata* (Fabricius). *Solenopsis richteri* and *S*. *invicta* have formed an extensive zone of hybridization in the USA [31], but hybridization has not been found in their native South America [32,33]. Colonies of *Solenopsis* spp. can contain >100,000 workers of varying size and are likely to be exposed to a variety of environmental stressors, including high temperature and low relative humidity, during their lifetimes, especially when foraging [12,34]. *Solenopsis richteri* occurs at slightly higher latitudes than *S*. *invicta* both in the USA and in their native South America (see maps in [35], pp. 27 and 59) [36,37], while *S*. *invicta* × *S*. *richteri* occur only in the USA between pure species populations. Furthermore, *S*. *invicta* and *S*. *geminata* are usually found in the tropical/subtropical USA, whereas *S*. *richteri* is usually found in more temperate areas in the USA [7,36]. The introduced *S*. *richteri* and *S*. *invicta* have each remained phenotypically constant since the time of their introduction from South America [38]. The potential range of expansion for *S*. *invicta*, based on current temperature and rainfall patterns, has been modeled for the continental USA [39] and worldwide [40]. *Solenopsis invicta* now occupies much of southern USA and has been established in California, West Indies, New Zealand, Australia, and parts of Asia [41,42,43,44].

A number of investigations have compared water use within and across some *Solenopsis* spp. and have identified a number of physiological traits related to water loss. Workers of *S*. *invicta* were more tolerant to desiccation stress than *S*. *richteri* workers despite the significantly greater body water content in *S*. *richteri* [45]. Size also plays a role in cuticular permeability in *Solenopsis* spp. For example, despite the greater percentage of total body water (%TBW) content of small *S*. *invicta* workers, cuticular permeability was greater in these individuals as compared with large *S*. *invicta* workers [12]. This has also been demonstrated in *S*. *xyloni* (McCook) workers [46]. Interestingly, Li and Heinz [47] indicated that desiccation resistance in polygyne *S*. *invicta* was not a function of body size and found low heritability of desiccation resistance in the tested population.

In addition to size, cuticular permeability was shown to be influenced by death from exposure to cyanide gas, which increased cuticular permeability in *S*. *invicta* workers by about 1.5-fold, and death by hexane extraction, which increased cuticular permeability by about 1.8-fold [12]. On the basis of a comparison among *S*. *invicta*, *S*. *richteri*, and *S*. *invicta* × *S*. *richteri*, Xu et al. [48] found that *Solenopsis* species with the highest water loss transition temperature (Tc) and highest melting point (Tm-max) of cuticular hydrocarbons (CHC’s) retained more water in relatively higher temperatures, and consequently were able to occupy warmer environments. Thus, several heritable physiological traits impact the water relations of *Solenopsis* spp. Similar impacts could also be caused by acclimation and environmental factors.

Studies within and across some *Solenopsis* spp. have revealed the degree of impact and relation of extreme abiotic factors to cutaneous water loss. Braulick et al. [49] compared lethal times (LT) in hours at 0% relative humidity (RH) and high temperatures for workers of *S*. *aurea* Wheeler, *S*. *geminata*, *S*. *invicta*, and *S*. *xyloni*. In general, workers survived progressively longer periods as temperature decreased; and major workers survived two to four times longer than minor workers at the same temperatures. Munroe et al. [50] compared the effect of desiccation on survival times in workers of *S*. *invicta*, *S*. *geminata*, and *S*. *xyloni*, and found that *S*. *invicta* had lower LT_50_ values than *S*. *geminata* and *S*. *xyloni.* Wendt and Verble-Pearson [51] found that major and medium *S*. *invicta* workers survived higher temperatures more often than did minor workers. Phillips et al. [52] found that workers of *S*. *invicta* from xeric conditions were less prone to desiccation than were those from moist conditions. It was suggested that this observation could be the result of natural selection at the population level or a physiological modification (acclimatization) by *S*. *invicta* workers as a consequence of continued exposure to more stressful environmental conditions. Li and Heinz [47] found that polygyne populations of *S*. *invicta* could be capable of adapting to arid habitats, therefore, suggesting an advantage in dominance of polygyne over monogyne *S*. *invicta* populations in arid habitats. Martin and Vinson [53] demonstrated that the ability to maintain a minimum viable level of body water could be a limiting factor to foraging range in *S*. *invicta* workers. Vogt et al. [54] also showed that temperature was a significant predictor of foraging activity in *S*. *invicta* workers. Thus, several environmental, as well as biological factors, impact the water relations of *Solenopsis* spp.

Although there have been several water relations studies on some *Solenopsis* species, there are no documented studies on the *S*. *invicta* × *S*. *richteri* hybrid, and there are none comparing multiple tropical and temperate *Solenopsis* species. Our objectives were to determine the percentage of total body water (%TBW) content, rate of mass loss, rate of total body water lost (%TBW lost), and cuticular permeability of introduced and native *Solenopsis* spp. of fire ants in the southeastern USA. In addition, we determined the desiccation sensitivity of these species. We hypothesized that tropical/subtropical fire ants (*S*. *invicta* and *S*. *geminata*) have lower cuticular permeability values and tolerate higher levels of desiccation (i.e., greater %TBW lost) than temperate fire ants (*S*. *richteri* and *S*. *invicta* × *S*. *richteri*). This study compares the water relations of temperate and tropical/subtropical fire ant workers across different latitudes in southern USA and is the first report of cuticular permeability and desiccation tolerance of *S*. *invicta* × *S*. *richteri* hybrid workers.

## 2. Materials and Methods

### 2.1. Study Species and Handling

The *Solenopsis invicta* workers were obtained from mounds on the Auburn University campus, Lee County, Alabama, in March 2017. The *Solenopsis richteri* workers were obtained from mounds in Hohenwald, Lewis County, and Mount Pleasant, Maury County, Tennessee, in March 2017. Workers of *S*. *invicta* × *S*. *richteri* hybrids were obtained from mounds in Cullman, Cullman County, and Hollywood, Jackson County, Alabama, in March 2017. Workers of *S*. *geminata* were obtained from mounds in Gainesville, Alachua County, Florida, in March 2017. GPS coordinates of ant collection locations are listed in Table 1. The identity of each species was confirmed by gas chromatography of hexane extracts of ca. 50 workers using both venom alkaloid and cuticular hydrocarbon characters, [32,55,56] following the methodology in Hu et al. [57]. Ants were used for experiments within the same week they were collected. On the basis of the large variation of body sizes and behavioral differences among size classes of *Solenopsis* spp. workers, each species was categorized, using the range of their head width, as small (0.72 mm or less), medium (0.73–0.92 mm), or large (0.93 mm or more) [58,59]. Sample size was either 14 or 15 individuals per worker size class per species. The sample size is indicated in Table 2.

We used live and Hydrogen Cyanide (HCN) killed ants in this study. Live ant data provide insight into desiccation tolerance, acceptable water loss for survival, and the abilities of the ants to actively regulate water loss, whereas dead insect data provide absolute cuticular permeability values. Live ants were separated by size and species and confined in a small glass jar with a screen lid. Jars were placed in a 1 L glass chamber with ≈10 g NaCN and KCN. HCN gas was generated by adding ≈0.5 mL hydrochloric acid to the cyanide salts. Ants were exposed for approximately 3 min or until all ants were dead. Dead ants were removed from the killing chamber and weighed to the nearest 0.01 mg individually using an electronic digital analytical balance.

### 2.2. Total Body Water Content

Knowledge of the rate of percentage of total body water (%TBW) lost and how much total water loss is lethal enabled the calculation of how long the ants would have to avoid or escape lethal environments. Total body water content of all ants was determined gravimetrically (see [12,60,61]). Masses of individual workers were measured to the nearest 0.01 mg on a digital balance in preweighed plastic weighing boats coated on the inside with Fluon^®^ to prevent escape of live ants. Then, ants were transferred to glass vials ringed on the upper inside surface with Fluon^®^ to prevent escape of live ants. Vials containing the ants were placed in an 11 L desiccating chamber containing approximately 0.5 kg anhydrous CaSO_4_ (Drierite^®^, W.A. Hammond Drierite Co. LTD, Xenia, OH, USA), resulting in 0–2% RH. Maximal effectiveness of the Drierite was ensured by heating it at 230 °C for at least 2 h prior to use to remove all water. The chamber was placed in an incubator and maintained at 30 ± 1 °C. Temperature and % RH in the chamber were monitored at each weighing, using a digital thermo-hygrometer. Specimens were weighed and returned to the desiccator chamber as quickly as possible. The ants were weighed at 0, 2, 4, 6, 8, 10, and 24 h; dried in a 55 °C oven for two days, weighed, and then dried an additional two days. Ants were dried and weighed until two successive weighings did not differ by >0.01 mg. Mass loss was assumed to be due entirely to water loss. The percentage of total body water (%TBW) was calculated as follows:%TBW content = [(M_initial_ − M_dry_)/M_initial_] * 100
where M_initial_ is the initial fresh body mass and M_dry_ is the dry mass.

### 2.3. Cuticular Permeability

Cuticular permeability was calculated from the difference between initial and 2 h desiccated masses. The mass loss after 2 h was used to avoid confounding factors caused by variability in body shape, and therefore surface area. Additionally, this period represented the maximum water gradient between the insect and the chamber, and thus the greatest water loss rate. Therefore, this period is the best estimate of absolute permeability [60].
$$\mathrm{Cuticular}\;\mathrm{permeability}=\frac{\mathrm{water}\;\mathrm{lost}\;\left(\mu g\right)}{\mathrm{Surface}\;\mathrm{area}\;\left({\mathrm{cm}}^{2}\right)*\mathrm{time}\;\left(h\right)*\mathrm{saturation}\;\mathrm{deficit}\;\left(\mathrm{mmHg}\right)}$$

Surface area (cm^2^) was estimated by Meeh’s formula. Surface area (cm^2^) = Initial mass (g)^2/3^ * 12 [62].

Saturation deficit, which is the difference between the vapor pressure of water at a given RH and temperature and the vapor pressure of saturated air at the same temperature [15], remained constant at 31.82 mmHg for 30 °C and 0% RH [63].

Adjusted mass loss was calculated as the difference between initial and 2 h masses divided by initial mass (g) and the formula is as follows:Adjusted mass loss = mg of H_2_O lost (T_0_ − T_2_)/gram initial body mass (T_0_)
where T_0_ is initial mass and T_2_ is mass after 2 h of desiccation.

### 2.4. Rates of Mass Loss, Water Loss, and Mortality

Hourly percentage of initial mean mass loss and %TBW loss of live and dead individuals of each species and size were plotted individually by time of desiccation; the analysis of these relationships is described below. The condition (live or dead) of live ants was recorded at each weighing.

### 2.5. Statistical Analysis

A randomized complete block design with body size, live/dead condition, and species as main effects and the size by species interaction was used to evaluate differences in cuticular permeability and %TBW among species within each size. Analysis of variance followed by the Tukey–Kramer HSD comparison test (*p* < 0.05; [64]) was performed on the initial mass, %TBW, and cuticular permeability data among and across the species. Linear regression was used to determine if cuticular permeability was related to initial live mass. Change in % mass loss or %TBW lost over time was analyzed using nonlinear regression [65]. A rectangular hyperbolic model was used for analysis of the change in % mass loss and %TBW lost overtime. The following function was fit to the hourly % mass loss and %TBW lost values:$$Y=\frac{ax}{b+x}$$
where *Y* = % mass loss or %TBW lost, *a* = the maximum asymptotic value of % mass loss or %TBW lost, *b* = (t max)/2 or the period required for half the maximum value to be reached, and *x* = hour. This function was selected because it is the most parsimonious expression that contains a curvilinear increase and an asymptotic maximum. The maximum value would be obtained with complete dryness of the specimen. For % mass loss, the maximum represents %TBW; for %TBW lost, the maximum should approximate 100%, or complete loss of all body water. Probit analysis [66] was used to estimate the median lethal time of ants exposed to desiccating conditions (ca. 30 °C and 0–2% RH). The resulting LT_50_ values were used to estimate desiccation tolerance, or %TBW lost at death, from the nonlinear regression equations. Data are expressed as means ± SE, α = 0.05.

## 3. Results

### 3.1. Body Mass and Water Content

Initial body masses of *S*. *invicta* workers ranged from 0.44–0.81 mg for small workers, 0.84–2.55 mg for medium workers, and 2.02–4.36 mg for large workers. Initial body masses of *S*. *richteri* workers ranged from 0.63–1.08 mg for small workers, 1.54–2.53 mg for medium workers, and 2.36–3.79 mg for large workers. Initial body masses for *S*. *invicta* × *S*. *richteri* workers ranged from 0.54–1.02 mg for small workers, 0.72–2.25 mg for medium workers, and 1.34–4.72 mg for large workers. Initial body masses for *S*. *geminata* workers ranged from 0.46–1.9 mg for small workers, 1.11–2.91 mg for medium workers, and 3.32–7.52 mg for large workers. Mean initial body masses for each size, and each species of live and dead ants are presented in Table 2.

#### 3.1.1. %TBW of Live Ants

The %TBW of small, medium, and large worker sizes was not significantly different among the four *Solenopsis* species (Figure 1A). Within each species, small workers had significantly greater %TBW than large workers (Appendix A). In all the four *Solenopsis* species, ranking of %TBW of the worker sizes was small > medium > large (Appendix A). There was no significant difference in %TBW between tropical/subtropical and temperate species (Figure 1C) when worker sizes were combined.

#### 3.1.2. %TBW of Dead Ants

Combining sizes, the %TBW was not significantly different between tropical/subtropical and temperate species (Figure 1B). When separated, the %TBW of small, medium, and large workers were not significantly different among the four *Solenopsis* species (Figure 1C). In each of the sizes, *S*. *richteri* had the lowest in %TBW, whereas small and large *S*. *invicta* had the highest in %TBW (Figure 1C). There were no significant differences in %TBW among the sizes in all four *Solenopsis* species (Appendix A).

#### 3.1.3. Adjusted Mass Loss of Live Ants

Temperate *S*. *richteri* had significantly greater adjusted mass loss than the tropical *S*. *invicta* (Figure 2A). There was no significant difference in adjusted mass loss among the four species in each size (Figure 2C). Small workers of *S*. *invicta* × *S*. *richteri* and *S*. *geminata* had significantly greater mass loss than large workers (Appendix A). Across all species, the ranking of adjusted mass loss was small > medium > large (Appendix A).

#### 3.1.4. Adjusted Mass Loss of Dead Ants

Combining sizes, temperate *S*. *invicta* × *S*. *richteri* had significantly greater adjusted mass loss, by ca. 1.3-fold, than subtropical *S*. *geminata* (Figure 2A). Adjusted mass loss of dead ants increased over that in live ants by ca. 1.2-fold and ca. 1.3-fold for *S*. *invicta* and *S*. *invicta* × *S*. *richteri* combined sizes, respectively (Figure 2A).

There were significant differences in adjusted mass loss between temperate (*S*. *richteri* and *S*. *invicta* × *S*. *richteri*) and subtropical (*S*. *geminata*) ant workers both in medium and large sizes (Figure 2B). Temperate *S*. *invicta* × *S*. *richteri* had significantly greater adjusted mass loss than tropical species *S*. *invicta* and *S*. *geminata* for both medium and large workers (Figure 2B). Ranking of species by adjusted mass loss of medium and large sizes was *S*. *invicta* × *S*. *richteri* > *S*. *richteri* > *S*. *invicta* > *S*. *geminata* (Figure 2B). There were significant differences in adjusted mass loss among the sizes within each species (Appendix A). Adjusted mass loss of small workers was greater than medium, and that of medium was greater than large workers in all species except *S*. *invicta* × *S*. *richteri* (Appendix A). Ranking of sizes by adjusted mass loss was small > medium > large for all the four *Solenopsis* species (Appendix A).

#### 3.1.5. Rates of % Initial Mass Loss and %TBW Lost

The % initial mass loss and %TBW lost increased as rectangular hyperbolic function of desiccation time (Appendix A); rising from an intercept of zero and increased at a declining rate eventually reaching an asymptote. The asymptote for the % initial mass loss curves approximate %TBW; those for %TBW lost approximate 100%. The rectangular hyperbolic function was appropriate as all the regressions were highly significant (*p* < 0.0001) with r^2^ values > 0.9 (Appendix A).

For live ants, the % initial mass loss (Appendix A) the maximum asymptotic value ranged from 70–107.9 for small *S. geminata* and large *S. invicta*, respectively. These asymptotic values (±2 SE) overlap the %TBW (Figure 1). The period required for half the maximum value to be reached ranged from 2.6 h for small *S. geminata* to 20.3 h for large *S. invicta* (Appendix A). Percentage TBW lost by live ants increased similarly (Appendix A) with the maximum asymptotic value overlapping 100% (representing 100% water loss) and the ”b” coefficient ranged from 2.4–20.2 h for small *S. geminata* and large *S. invicta*, respectively (Appendix A). The ”b” coefficients for both % initial mass loss and %TBW lost increased with increasing ant size for all species.

For dead ants, % initial mass loss (Appendix A) the ”a” coefficient or maximal asymptotic values were slightly lower than for live ants, ranging from 67.2% for small *S. richteri* to 97.2% for large *S. geminata* (Appendix A). These values were similar to those for %TBW (Appendix A). In all the four *Solenopsis* species, the ”b” coefficient or period required for half the maximum value to be reached ranged from 1.7–16.2 h for small and large *S. geminata*, respectively (Appendix A). Appendix A illustrates the relationship between %TBW lost by dead ants and desiccation time. The maximum asymptotic value or ”a” coefficient overlaps or exceeds 100% (representing complete dryness). The period required for half the maximum value to be reached or ”b” coefficient ranged from 1.6–16.2 h for small and large *S. geminata*, respectively (Appendix A). The ”b” coefficients for both % initial mass loss and %TBW lost increased with increasing ant size for all species.

### 3.2. Cuticular Permeability

Combining sizes, cuticular permeability of dead ants was significantly greater in *S*. *richteri* and *S*. *invicta* × *S*. *richteri* workers than in *S*. *geminata* workers (*p* < 0.05) (Figure 3A). Cuticular permeability was about 1.3-fold greater in temperate *S*. *invicta* × *S*. *richteri* than in subtropical *S*. *geminata* workers; and about 1.2-fold greater in temperate *S*. *richteri* than in subtropical *S*. *geminata* workers (Figure 3A). Cuticular permeability values of *S*. *richteri* and *S*. *geminata* were not significantly different from that of *S*. *invicta* (Appendix A). Among temperate ants, cuticular permeability was significantly greater in *S*. *invicta* × *S*. *richteri* than *S*. *richteri* (Figure 3A). Calculated cuticular permeability of live *S*. *richteri* ants was significantly greater than that of *S*. *invicta* (Figure 3A).

There was no significant difference in cuticular permeability among small workers of the four *Solenopsis* species (Figure 3B). Medium *S*. *invicta* × *S*. *richteri* workers had significantly greater cuticular permeability than tropical/subtropical *S*. *invicta* and *S*. *geminata* workers, whereas tropical/subtropical *S*. *invicta* and *S*. *geminata* worker ants had similar cuticular permeability values (Figure 3B). Large temperate *S*. *richteri* and *S*. *invicta* × *S*. *richteri* workers had significantly greater cuticular permeability than tropical *S*. *geminata* (Appendix A). Large temperate *S*. *invicta* × *S*. *richteri* workers had significantly greater cuticular permeability than tropical/subtropical *S*. *invicta* and *S*. *geminata* workers (Figure 3B). There were no significant differences in cuticular permeability among sizes for *S*. *invicta*, *S*. *richteri*, and *S*. *invicta* × *S*. *richteri* (Figure 3C). However, small workers had significantly greater cuticular permeability than both medium and large workers in *S*. *geminata* (Figure 3C).

### 3.3. Desiccation Tolerance

The LT_50_ values (h) for all live size classes of *S*. *richteri*, *S*. *invicta* × *S*. *richteri*, *S*. *invicta*, and *S*. *geminata* worker fire ants are shown in Table 3. All probit analyses were significant (*p* < 0.05) and the LT_50_ values ranged from 1.5 h for small *S*. *geminata* workers to 8.5 h for large *S*. *invicta* workers. The LT_50_ values increased with increasing ant size. Desiccation tolerance estimated as the percentage of total body water (%TBW) lost at the LT_50_ time is shown in Table 4. Median %TBW lost at death ranged from 35.7% for small *S. geminata* to 49.8% for large *S. invicta*. There were no significant differences in %TBW lost at death, based on overlap of the 95% confidence intervals, among the size classes of any of the *Solenopsis* spp. The mean %TBW lost, at the LT_50_ time, ranged from 42.5% for *S. geminata* to 45.8% for *S. invicta*.

## 4. Discussion

The data support our hypothesis that tropical/subtropical fire ants (*S*. *invicta* and *S*. *geminata*) have lower cuticular permeability values than temperate fire ants (*S*. *richteri* and *S*. *invicta* × *S*. *richteri*). However, the data do not support the general hypothesis that tropical/subtropical fire ants can tolerate greater levels of desiccation (i.e., greater % total body water lost). Comparing cuticular permeability values among dead ants (i.e., absolute permeability), tropical *S*. *invicta* and *S*. *geminata* had significantly lower cuticular permeability values than that of temperate *S*. *invicta* × *S*. *richteri*. In addition, cuticular permeability values of dead *S*. *geminata* were significantly lower than those of *S*. *richteri* and *S*. *invicta* × *S*. *richteri*. Cuticular permeability values obtained for dead *S*. *invicta* were similar to those reported by Appel et al. [12]. Our data also suggest that large live *S*. *invicta* are significantly less vulnerable to desiccation stress than large live *S*. *richteri*. All these results partially explain the predominant location of each of these species, as *S*. *invicta* are found in lower latitudes and tropical/subtropical regions, whereas *S*. *richteri* are found in higher latitudes and more temperate regions in their native South America. In addition, the climate in their current distribution in the southeast USA is similar to that of their native distribution. This suggests that cuticular permeability and desiccation tolerance play a role in the distribution and adaptation of invasive ants. Considering the ranges of cuticular permeability values that we obtained, neither of the tropical/subtropical *S*. *invicta* nor *S*. *geminata* workers fell within the range for temperate *S*. *invicta* × *S*. *richteri* workers. This suggests that differences in habitat preference could exist among these species.

The body water content of 67.56 ± 0.90% for combined sizes of *S*. *invicta* in our data is similar to the 68.1% reported by Elzen [67] and the 63% by Appel et al. [12]. The body water contents of *S*. *invicta* and *S*. *richteri* have been reported to be significantly different [45]. However, our data show no body water content difference for combined sizes of workers of both the tropical/subtropical species (*S*. *invicta* and *S*. *geminata*) as compared with the temperate species (*S*. *richteri* and *S*. *invicta* × *S*. *richteri*). The higher number of replicates in the study by Chen et al. [45] could have contributed to the detection of significant differences in their study. The body water content values recorded for all four *Solenopsis* spp. in the current study are similar to those reported for workers of other ant species, including the desert ant *Pogonomyrmex rugosus* (Emery) (65.9%), *P*. *occidentalis* (Cresson) (63.4%), and *Messor pergandei* (Mayr) (64.7%) (see references in [14] (Table 2.1)). It is possible that temperate and tropical/subtropical fire ants are similar in body water content. However, a comparison based on the present data could be insufficient.

Large workers had significantly lower percent total body water (%TBW) lost values than small workers in all tested species. This is similar to results by Appel et al. [12], where *S*. *invicta* small workers contained significantly more body water (*p* < 0.05) than large workers. Additionally, the values of %TBW in small *S*. *invicta* workers, in our study (72.69 ± 1.65), was close to that by Appel et al. [12] (65.02 ± 1.30). The %TBW values in large *S*. *invicta* workers, in our study (63.93 ± 1.03), was also similar to that by Appel et al. [12] (61.07 ± 0.49). In the present study, %TBW was approximately 1.14-fold lower in large than in small *S*. *invicta* workers; 1.13-fold lower in large than in small *S*. *geminata* workers; 1.11-fold lower in large than in small *S*. *invicta* × *S*. *richteri* workers; and 1.08-fold lower in large than in small *S*. *richteri* workers. Combining stages, the %TBW was not significantly different among workers of the four species. Similarly, there was no significant difference in %TBW between *S*. *invicta* and *S*. *richteri* female alates but it was significantly lower in workers of *S*. *invicta* than in *S*. *richteri* [45]. The higher average %TBW for small workers could be due to the behavior of this stage in the colony. Small workers are more likely to care for broods than are other castes [58]. Brood care in ants requires the availability of liquid food for trophalaxis, therefore, samples of small workers could contain individuals with full and empty crops. In addition, small workers that forage are more likely to forage for liquid food [12,32].

Cuticular permeability values of dead ants give absolute comparisons to other arthropods. The cuticular permeability values obtained, in the current study, for *S*. *invicta* workers were small (33.03 µg cm^−2^ h^−1^ mmHg^−1^), medium (30.26 µg cm^−2^ h^−1^ mmHg^−^^1^), and large (26.54 µg cm^−2^ h^−1^ mmHg^−1^). These values are similar to those reported for other arthropods from more or less tropical habitats, such as *Hadrurus hirsutus* (Wood); Scorpiones = 25 µg cm^−2^ h^−1^ mmHg^−1^, *Locusta migratoria* (L.); Orthoptera = 22 µg cm^−2^ h^−1^ mmHg^−1^ and *Hemilepistus reaumuri* (Milne-Edwards); and Isopods = 23 µg cm^−2^ h^−1^ mmHg^−1^. Cuticular permeability values for *S*. *geminata* workers (small = 35.14, medium = 27.97, and large = 23.34 µg cm^−2^ h^−1^ mmHg^−1^) are also similar to those measured in arthropods of more or less tropical habitats, such as *Venezillo arizonicus* (Mulaik and Mulaik); Isopods = 32 µg cm^−2^ h^−1^ mmHg^−1^. Cuticular permeability values for *S*. *richteri* workers (small = 36.46, medium = 33.70, and large = 29.37 µg cm^−2^ h^−1^ mmHg^−1^) are similar to those measured in arthropods of temperate habitats, such as *Lycosa amentata* (Clerck); Araneae = 28.3 µg cm^−2^ h^−1^ mmHg^−1^. Similarly, cuticular permeability values for *S*. *invicta* × *S*. *richteri* workers (small = 38.03, medium = 40.99, and large = 36.20 µg cm^−2^ h^−1^ mmHg^−1^) are similar to those measured in insects of temperate habitats, such as *Chortoicetes terminifera* (Walker); Orthoptera = 41 µg cm^−2^ h^−1^ mmHg^−1^ (see references in [15] (Table 6)). It should be noted that the designation of arthropods mentioned above based on their cuticular permeability values alone are meant to be more of a guide than an absolute designation. Although there is an overlap in the range in cuticular permeability values of small workers between tropical (*S*. *invicta* and *S*. *geminata*) and temperate (*S*. *richteri* and *S*. *invicta* × *S*. *richteri*) species in our study, there are no overlaps in medium and large workers of these categories of species. Workers of all sizes engage in foraging. Small workers with head widths around 0.7 mm often tend broods, scout, and recruit large workers to forage food [37,68,69]. Furthermore, these data suggest that cuticular permeability of tropical/subtropical fire ants, in our study, exhibit a degree of adaptation to more or less tropical environments, whereas the cuticular permeability of temperate fire ants is more similar to arthropods adapted to temperate environments [40].

Cuticular permeability values of dead workers (averaging across the three size classes) of the four species ranked as follows: *S*. *invicta* × *S*. *richteri* > *S*. *richteri* > *S*. *invicta* > *S*. *geminata*. Similarly, the cuticular permeability values of dead large workers ranked as follows: *S*. *invicta* × *S*. *richteri* > *S*. *richteri* > *S*. *invicta* > *S*. *geminata*. This is similar to the ranking of initial mass of large workers. Cuticular permeability and initial mass tended to follow a similar trend in large workers of all species in our results. Thus, given the cuticular permeability of these *Solenopsis* species, a relatively larger rather than small body size could better tolerate water loss if water loss becomes a limiting factor when foraging. This result is similar to that of foraging harvester ants, *Pogonomyrmex rugosus* (Emery) [70]. Meeh’s formula was used to estimate surface area in our study to calculate cuticular permeability because body size affects water loss (surface area to volume ratio). Linear regressions of cuticular permeability on body mass were performed to determine if Meeh’s formula had adequately compensated for the relationship between mass and surface area. The majority (>90%) of these regressions were not significant, indicating that Meeh’s formula provided a reasonable estimate for surface area.

In the present study, there was no significant difference between the absolute cuticular permeability values of small and large workers among the four *Solenopsis* species (except in *S*. *geminata*). However, a significant difference was detected between the cuticular permeability values of small and medium workers of *S*. *invicta* when more replicates were used by Appel et al. [12]. We speculate that significantly different results could have been detected in our study if more replicates were used, and if this were the case, the lower cuticular permeability of large workers could be related to the foraging behavior of this stage in the colony.

An adjusted mass loss was used as a way to compare water loss from live and dead ants without making assumptions about the source of that water loss which could be through respiration, secretions, and feces, in addition to the cuticle. If adjusted mass loss were greater for live as compared with dead insects, and if cuticular permeability was the opposite, then differences would be due to respiratory, fecal, or secretory water loss. However, if adjusted mass loss was similar for live and dead insects, then it could be argued that these other routes of water loss would be insignificant as compared with cuticular water loss. Combining worker sizes, adjusted mass loss was similar between live and dead ants (Figure 2A). Mean values were slightly greater in dead *S*. *invicta* × *S*. *richteri* and *S. invicta*. Thus, we speculate that these species could have more active cuticular water “pumps” [24,25].

Our study also attempted to relate the desiccation tolerance of four species of *Solenopsis* fire ants with their geographic distributions. The ranking of the LT_50_ values of large workers in our results was *S*. *invicta* > *S*. *geminata* > *S*. *invicta* × *S*. *richteri* > *S*. *richteri* (Table 3). Large *S. invicta* workers had LT_50_ values approximately 2.3 times greater than those of large *S*. *richteri*. Large workers take part in foraging [58], thus our LT_50_ data suggests similarity in ranking of desiccation tolerance among these four species. This indicates that *S*. *invicta* large workers are more desiccation tolerant than *S*. *geminata*, *S*. *invicta* × *S*. *richteri*, and *S*. *richteri* large workers. Our data suggest that foraging large *S*. *invicta* workers could tolerate exposure to desiccating conditions significantly longer than large *S*. *richteri* workers. Thus, *S*. *invicta* is expected to be able to forage in areas of high insolation longer than the other three species, perhaps contributing to their relative distributions [71]. Temperate fire ants (*S*. *invicta* × *S*. *richteri* and *S*. *richteri*) have a lower LT_50_ than tropical and subtropical fire ants (*S*. *invicta* and *S*. *geminata*); this suggests that the ants could have more difficulty than tropical/subtropical fire ants surviving drier environmental conditions than those of their present range. In general, live large fire ants desiccate at a slower rate than live small fire ants. For instance, the increases in LT_50_ values for large workers as compared with small workers were 5.23, 3.42, 2.58, and 1.65 times, in *S*. *geminata*, *S*. *invicta*, *S*. *invicta* × *S*. *richteri,* and *S*. *richteri*, respectively. The desiccation rates between small and large fire ants in our results are similar to those reported by Munroe et al. [50]. These results supported the suggestion by Edney [15] that, in moisture-deficient situations, the amount of water loss in terms of total body water initially present was greater in small animals than large animals. The ranking of the LT_50_ values of large workers, in our results, was opposite to that of mass loss among the four live species, *S*. *richteri* > *S*. *invicta* × *S*. *richteri* > *S*. *geminata* > *S*. *invicta*. Our results suggest that the reason large workers are able to carry a greater proportion of foraging responsibilities is that they are desiccation resistant and tolerant [50,72].

The composition of epicuticular lipids enable tropical *S*. *invicta* to better cope with desiccation than temperate *S*. *richteri* [45,48]. Our data showed that *S*. *invicta* can tolerate desiccation more than *S*. *invicta* × *S*. *richteri* and *S*. *richteri*. This agrees with the findings of Xu et al. [48] in that the highest melting points of samples of cuticular hydrocarbons (CHCs) from *S*. *invicta* and *S*. *invicta* × *S*. *richteri* were significantly higher than that from *S*. *richteri*. Cuticular hydrocarbon profiles of *S*. *richteri* are characterized by significant amounts of short-chain (C_23_–C_27_) saturated and unsaturated hydrocarbons. In contrast, profiles of *S*. *invicta* consist primarily of long-chain (C_27_–C_29_) saturated hydrocarbons; unsaturated alkenes are completely lacking. The hybrid *S*. *invicta* × *S*. *richteri* shows intermediate profiles of the two parent species [48]. Long-chain saturated waxy hydrocarbons are better at water proofing and have higher melting points than shorter chain unsaturated chains.

Although *S*. *invicta* and *S*. *geminata* inhabit incredibly humid locales such as the Gulf Coast states in the USA, the task of foraging carried out by workers of these species could expose them to highly desiccating conditions, in contrast to the relatively cooler and more humid conditions in which temperate fire ant species abundantly inhabit. Tropical/subtropical fire ants can also benefit from the relatively cool and humid conditions in the deep parts of ant nests, where cuticular water loss is likely to be minimal [70]. As a result of these adaptations, we infer that the water loss rate and desiccation tolerance could be part of a complex of physiological and behavioral factors behind the distribution of these terrestrial insects [70]. Queen ants of different species vary in their desiccation tolerance [73], and if this were the case among *Solenopsis* species as well, then we would expect queens of tropical *S*. *invicta* to be more desiccation tolerant than the queens of temperate *S*. *richteri*. Thus, this difference would play a role in colony establishment and limiting distribution among these species. Nevertheless, water vapor pressure deficits are likely to be low for claustral colony founder queens that are found deep in the nests [70].

Large workers engage in foraging in fire ant colonies [58]. Thus, they are exposed to potentially desiccating conditions. It is possible that there are qualitative and quantitative differences in the cuticular hydrocarbon profiles of small and large workers of these four *Solenopsis* species, as was found in *S*. *saevissima* [74]. Foraging in ants tends to require the capacity to tolerate desiccating conditions [71], thus, samples of large workers can include individuals with low cuticular permeability, enabling them to tolerate desiccating conditions better than non-frequent foragers such as the small workers. If this were the case, then it would be expected that more resources would be invested in making the waxes on large workers that are more frequent foragers than small workers that engage more often in brood care within the nest [58]. We speculate that small workers possess greasier cuticular hydrocarbons while large workers have waxier cuticular hydrocarbons. Future studies should investigate if small workers have enough wax to not dry out immediately, and enough grease to readily sense pheromones among other semiochemicals. Thus, this could indicate if there exists a tradeoff between waterproofing wax and grease that allows better chemoreception.

## 5. Conclusions

In conclusion, this study illustrates differences in body mass, %TBW, cuticular permeability, and desiccation tolerance of *S*. *invicta*, *S*. *richteri*, *S*. *invicta* × *S*. *richteri*, and *S*. *geminata*, which are important pests in the southeastern U.S. There are differences in the water relations of species adapted to temperate as compared with tropical latitudes. Temperate *S*. *invicta* × *S*. *richteri* had significantly greater cuticular permeability than the tropical *S*. *invicta* and *S*. *geminata*. Live temperate *S*. *richteri* lost significantly more %TBW than tropical *S*. *invicta*. These results provide insights into differences in the water relations of these four *Solenopsis* species and help to explain the relative distribution of these species in the southeastern USA. The capability of these four *Solenopsis* species to survive, limit cuticular water loss, and tolerate desiccation influences their distribution both in their native South America and in their introduced North American range. Extremes of hot and cold temperature and low relative humidity limit the range of *S*. *invicta* and *S*. *richteri* [7,40,50,75,76,77]. However, during global warming, the ranges of *S*. *invicta* and *S*. *richteri* are predicted to increase to the north of their present range [40], and *S. invicta* to the west and east of its present range [7,50]. There is evidence in the literature that fire ants engage in several behaviors in response to stressful abiotic conditions. These include burying food resources for foraging during hotter parts of the day, foraging at night when it is cool, and creating extensive underground tunnels. Further studies including physiological and genetic analyses of desiccation tolerance in *Solenopsis* spp. are needed to determine what physiological and genetic attributes enable the tropical *S*. *invicta* to tolerate desiccation better than the temperate *S*. *richteri*.

## Figures and Tables

**Figure 1 insects-11-00418-f001:**
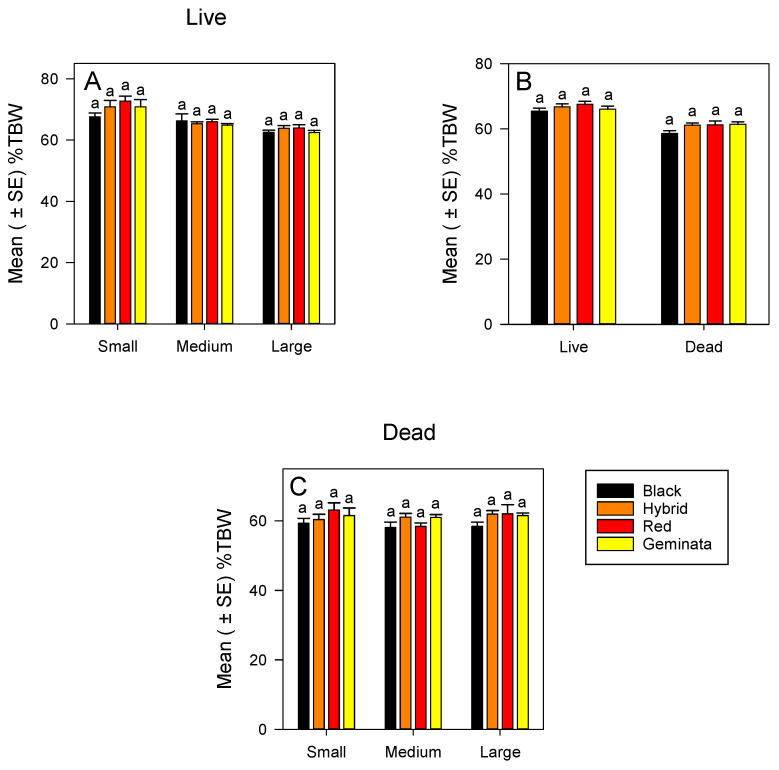
Mean (± SE) percentage of total body water (%TBW) for: (**A**) Live *S*. *richteri* (Black), *S*. *invicta* × *S*. *richteri* (Hybrid), *S*. *invicta* (Red), and *S*. *geminata* (Geminata) fire ants compared within each worker size class; (**B**) Combined worker size classes of *S*. *richteri*, *S*. *invicta* × *S*. *richteri*, *S*. *invicta*, and *S*. *geminata* fire ants compared within each live/dead status; and (**C**) Dead *S*. *richteri*, *S*. *invicta* × *S*. *richteri*, *S*. *invicta*, and *S*. *geminata* fire ants compared within each worker size class. Means with the same letter within each worker size class or species are not significantly different (*p* < 0.05). *n* = 14 or 15 individuals per worker size class per species.

**Figure 2 insects-11-00418-f002:**
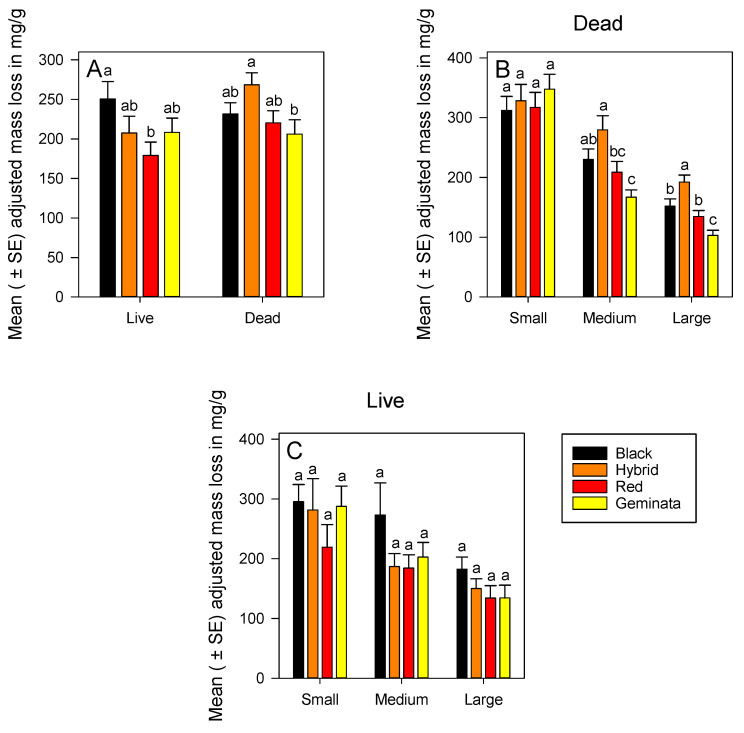
Mean (± SE) adjusted mass loss for: (**A**) Combined worker size classes of *S*. *richteri* (Black), *S*. *invicta* × *S*. *richteri* (Hybrid), *S*. *invicta* (Red), and *S*. *geminata* (Geminata) fire ants compared within each live/dead status; (**B**) Dead *S*. *richteri*, *S*. *invicta* × *S*. *richteri*, *S*. *invicta*, and *S*. *geminata* fire ants compared within each worker size class; and (**C**) Live *S*. *richteri*, *S*. *invicta* × *S*. *richteri*, *S*. *invicta*, and *S*. *geminata* fire ants compared within each worker size class. Means with the same letter within each worker size class or species are not significantly different (*p* < 0.05). *n* = 14 or 15 individuals per worker size class per species.

**Figure 3 insects-11-00418-f003:**
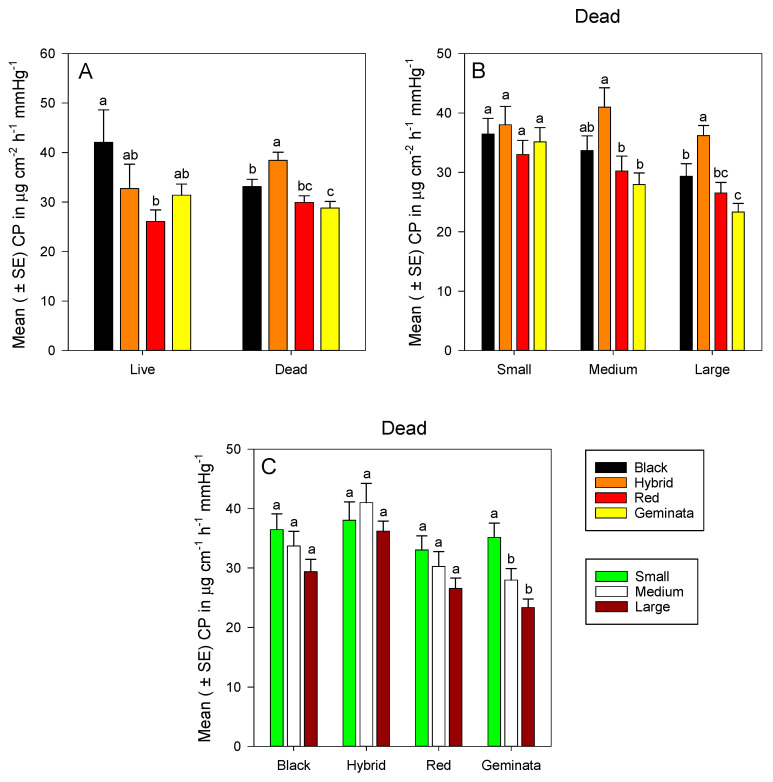
Mean (± SE) calculated cuticular permeability (CP) for: (**A**) Combined worker size classes of *S*. *richteri* (Black), *S*. *invicta* × *S*. *richteri* (Hybrid), *S*. *invicta* (Red), and *S*. *geminata* (Geminata) fire ants compared within each live/dead status; (**B**) Dead *S*. *richteri*, *S*. *invicta* × *S*. *richteri*, *S*. *invicta*, and *S*. *geminata* fire ants compared within each worker size class; and (**C**) Dead small, medium, and large ant worker size classes compared within each *Solenopsis* species.

**Table 1 insects-11-00418-t001:** Collection locations of *S*. *invicta*, *S*. *richteri*, *S*. *invicta* × *S*. *richteri,* and *S*. *geminata* fire ants in the USA.

Species	Colony	Town/City, State	Latitude	Longitude
*Solenopsis geminata*	1	Gainesville, FL	29°34’19.49” N	82°27’22.89” W
	2	Gainesville, FL	29°34’24.92” N	82°27’30.49” W
	3	Gainesville, FL	29°34’25.23” N	82°27’29.59” W
*S*. *invicta*	1	Auburn, AL	32°37’31.47” N	85°30’07.78” W
	2	Auburn, AL	32°34’36.73” N	85°29’52.24” W
	3	Auburn, AL	32°36’59.58” N	85°30’27.14” W
*S*. *richteri*	1	Waverly, TN	36°05’01.41” N	87°48’31.32” W
	2	Hohenwald, TN	35°33’24.66” N	87°31’46.43” W
	3	Mount Pleasant, TN	35°36’38.52” N	87°15’46.40” W
*S*. *invicta* × *S*. *richteri*	1	Huntsville, AL	34°32’18.66” N	86°30’00.27” W
	2	Huntsville, AL	34°34’45.70” N	86°33’13.75” W
	3	Decatur, AL	34°31’47.41” N	86°54’02.96” W

**Table 2 insects-11-00418-t002:** Initial mass (mg) of small, medium, and large live and dead workers of *S*. *invicta*, *S*. *richteri*, *S*. *invicta* × *S*. *richteri,* and *S*. *geminata* ants (mean ± SE).

Species	Size Class	N for Live Ants	Initial Mass (mg) Live Ants	N for Dead Ants	Initial Mass (mg) Dead Ants
*S*. *richteri*	Small	15	0.89 ± 0.03 **A	15	0.72 ± 0.03 cA
	Medium	15	2.94 ± 0.03 *A	15	1.47 ± 0.09 bB
	Large	15	3.19 ± 0.12 *B	15	3.31 ± 0.15 aB
*S*. *invicta* × *S*. *richteri*	Small	15	1.18 ± 0.42 **A	15	0.70 ± 0.02 cA
	Medium	15	1.71 ± 0.16 **A	15	1.49 ± 0.11 bB
	Large	14	3.08 ± 0.27 *B	14	3.21 ± 0.26 aB
*S*. *invicta*	Small	15	0.59 ± 0.02 ***A	15	0.52 ± 0.02 cB
	Medium	14	1.61 ± 0.13 **A	15	1.41 ± 0.09 bB
	Large	15	3.23 ± 0.19 *B	15	3.52 ± 0.14 aB
*S*. *geminata*	Small	15	0.63 ± 0.09 ***A	15	0.47 ± 0.02 cB
	Medium	15	2.07 ± 0.14 **A	15	2.23 ± 0.19 bA
	Large	15	5.26 ± 0.35 *A	15	5.63 ± 0.37 aA

Means of the three size classes within a species followed by the same lowercase letter (for dead ants) or number of asterisks (for live ants) are not significantly different (*p* < 0.05). Means of the four species, live or dead ants, within a size class followed by the same uppercase letter are not significantly different (*p* < 0.05). N = numbers of individuals.

**Table 3 insects-11-00418-t003:** LT_50_ values (h) for live small, medium, and large workers of *S*. *richteri*, *S*. *invicta* × *S*. *richteri*, *S*. *invicta*, and *S*. *geminata* ants.

Species	Size	N	LT_50_ (h)	(95% CI)	Slope ± SE	χ^2^	**Df**
*S*. *richteri*	Small	15	2.21	(1.59–2.71)	6.14 ± 1.75	0.09	4
	Medium	15	3.03	(2.19–3.76)	4.07 ± 0.84	1.44	4
	Large	15	3.65	(2.57–4.62)	3.20 ± 0.67	2.20	4
*S*. *invicta* × *S*. *richteri*	Small	15	2.08	(1.36–2.58)	5.65 ± 1.69	0.12	4
	Medium	15	3.63	(2.87–4.34)	5.02 ± 0.96	1.77	4
	Large	14	5.38	(4.24–6.67)	3.50 ± 0.68	3.76	4
*S*. *invicta*	Small	15	2.49	(1.44–3.29)	3.21 ± 0.76	0.60	4
	Medium	14	4.50	(3.61–5.35)	4.81 ± 0.91	0.93	4
	Large	15	8.52	(6.14–13.81)	3.38 ± 0.67	4.29	4
*S*. *geminata*	Small	15	1.45	(0.27–2.22)	3.06 ± 1.03	1.08	4
	Medium	15	4.15	(3.28–4.99)	4.44 ± 0.85	2.57	4
	Large	15	7.59	(6.11–9.79)	3.35 ± 0.67	2.66	4

N, numbers of individuals; CI, confidence interval.

**Table 4 insects-11-00418-t004:** Percentage of total body water (%TBW) lost at median time of death; at lower and upper confidence intervals (CI) for live small, medium, and large workers of *S*. *richteri*, *S*. *invicta* × *S*. *richteri*, *S*. *invicta*, and *S*. *geminata* desiccated at 30 °C and 0–2% RH.

Species	Size	N	%TBW Lost at LT_50_	%TBW Lost at Lower CI	%TBW Lost at Upper CI	Mean %TBW Lost at LT_50_ for All Sizes in a Species ^a^
*S*. *richteri*	Small	15	48.05	39.49	53.53	45.42
	Medium	15	48.69	39.93	54.77	
	Large	15	39.53	30.56	46.33	
*S*. *invicta* × *S*. *richteri*	Small	15	42.64	27.88	48.27	43.31
	Medium	15	44.57	38.12	49.76	
	Large	14	42.73	35.82	49.61	
*S*. *invicta*	Small	15	40.97	28.07	48.39	45.79
	Medium	14	46.59	40.29	51.50	
	Large	15	49.80	39.13	68.17	
*S*. *geminata*	Small	15	35.74	9.35	46.05	42.47
	Medium	15	44.91	38.47	50.24	
	Large	15	46.76	39.90	55.63	

N = numbers of individuals; ^a^ mean of the three sizes.

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
