# Peer review of "Comparative Cutaneous Water Loss and Desiccation Tolerance of Four Solenopsis spp. (Hymenoptera: Formicidae) in the Southeastern United States"

_insects, 2020, doi:10.3390/insects11070418_

Round 1
Reviewer 1 Report
The authors quantified a wide range of different physiological factors in four solenopsis species to explain their distribution patterns. Unfortunately, the paper lacks clarity and is cluttered with irrelevant and/or insignificant results, which distract from the main message. I recommend the authors to cut the length of the manuscript at least by half and remove most figure panels and tables at least into the supplementary material.
Title: Shorten the title significantly, by at least half. It is unnecessarily long and convoluted.
L13: Start your Abstract with a broader basic introduction to the field before mentioning what data you collected.
L17: what does RH stand for? Also the sentence is not clear. When did you dry the ants in the oven? After a set time? After they died?
L19: Try not to use sentence structures with respectively. They break reading flow and are much harder to understand.
L20: What does LT50 stand for? The 1.5 is missing its unit.
L35-36: The sentence is unclear.
I don’t understand why you would expect tropical species living in a more humid environment, not adapted to dry climate, to have greater desiccation tolerance than temperate species?
What was the sample size for each experiment? It does not appear in the Methods section. Was only one sample collected per location?
Concerning Table 2: What was the sample size per colony? For your total N=15 I assume you had 5 per colony? If so did you consider your samples as pseudoreplicates by calculating the mean as the total 15 or did you previously calculate the mean first for each colony respectively and then calculated your initial mass? (in which case true sample size would be N=3). I would advice to keep the statistical results for different comparisons in separate columns for clearer presentation (after the values), at the moment it is very hard to clearly see what is significant.
L225 The legend should be self explanatory, the panels are not defined in the legend.
As far as I can tell, only Fig. 1B holds significant information with an actual relevant statistical result, justifying its graphical representation in the main manuscript. In my opinion, there is no need for the other graphs in the main manuscript or even in the supplemental Material, they could just as well be summarized in the text without a full page Figure.
Moreover, in Figure 1 and 2 it is unclear to me what exactly the posthoc test contains. Are you comparing all results across a figure panel? Or just between subsets of the panel? (ie only between size classes, alive or dead, or across species)
It is also unclear in both graphs what the sample size for each bar plot was.
Figure 3 Figure panels are missing. It is also unclear what the actual sample size is for each data point (with variance).
Why is there only one data point for large and medium geminata in the fourth panel of Fig. 3? The same question also in Figure 4, 5 and 6.
Tables 3-8 Include sample sizes as a column.
L395-406: Please improve on readability. A sentence which is 12 lines long with multiple parentheses and side sentences is impossible to read clearly! There is also no need to keep repeating detailed results in the discussion.
Author Response
Reviewer #1
Title: Shorten the title significantly, by at least half. It is unnecessarily long and convoluted.
Authors’ response: We have shortened it to “Comparative Cutaneous Water Loss and Desiccation Tolerance of Four Solenopsis spp. (Hymenoptera: Formicidae) in the Southeastern United States”
L13: Start your Abstract with a broader basic introduction to the field before mentioning what data you collected.
Authors’ response: We have started the Abstract with the following broader basic introduction before mentioning what we collected: “The high surface area to volume ratio of terrestrial insects makes them highly susceptible to desiccation mainly through the cuticle. Cuticular permeability (CP) is usually the most important factor limiting water loss in terrestrial insects. …”
L17: What does RH stand for? Also, the sentence is not clear. When did you dry the ants in the oven? After a set time? After they died?
Authors’ response: RH stands for ‘Relative Humidity’, and we have stated the full phrase in the Abstract. We have rephrased the sentence in Line 17 to: “Workers were periodically weighed and quickly returned into a desiccation chamber (30 ± 1 °C and 0 – 2 % Relative humidity). After 24 hours, they were dried in a 55 °C oven for two days. …”
L19: Try not to use sentence structures with “respectively”. They break reading flow and are much harder to understand.
Authors’ response: We have changed Line 19 to: “Solenopsis geminata had 1.3-fold lower CP value than S. invicta × S. richteri, and 1.2-fold lower CP value than S. richteri.”
L20: What does LT50 stand for? The 1.5 is missing its unit.
Authors’ response: LT50 stands for the lethal time to kill 50% of the population, and this has been stated in the Abstract. We have added the missing unit (h) to 1.5.
L35-36: The sentence is unclear.
Authors’ response: We have changed Lines 35 – 36 to: “Desiccation tolerance is limited by apparent morphological and ecological factors.”
I do not understand why you would expect tropical species living in a more humid environment, not adapted to dry climate, to have greater desiccation tolerance than temperate species?
Authors’ response: There are two components of ‘tropical’ habitats: heat and humidity. Solenopsis invicta comes from hotter area than S. richteri, both species live in humid areas. The greater temperature alone is expected to support our hypothesis.
Furthermore, (1) Previous studies such as Chen et al. (2014) indicated that S. invicta is more tolerant to heat and desiccation stress than S. richteri.
(2) Due to the high surface area to volume ratio of terrestrial insects, cuticular permeability and/or desiccation tolerance are important physiological traits impacting the adaptation of terrestrial species to tropical or temperate habitats.
(3) In their native land, S. invicta and S. geminata are predominantly found in tropical and subtropical habitats; while S. richteri is predominantly found in somewhat more temperate habitats (Trager 1991, Ometto et al. 2012). Furthermore, it is expected that there would be a relatively greater interspecific divergence in the patterns of selection acting on workers of these species, because adult workers most directly experience the distinct environments characterizing the different ranges occupied by S. invicta and S. geminata (namely, tropical and subtropical habitats) and S. richteri (somewhat more temperate habitats) (Trager 1991, Ometto et al. 2012).
Should climate change lead to reduced humidity within tropical habitats, S. invicta and S. geminata abundances and the rates of the ecosystem roles they perform could be reduced, however not as reduced as those of S. richteri and S. invicta × S. richteri.
Humid environment will provide an advantage for tropical terrestrial species to live in, compared to arid environment. Thus, desiccation may not be a huge concern in such habitat. However, when exposed to humid habitats, there is higher expectation that tropical species would adapt better than non-tropical species.
What was the sample size for each experiment? It does not appear in the Methods section. Was only one sample collected per location?
Authors’ response: We have included the following not the Materials and Methods section: “Sample size was either 14 or 15 per worker size per species. The sample size is indicated in Table 2”
Concerning Table 2: What was the sample size per colony? For your total N=15, I assume you had 5 per colony? If so, did you consider your samples as pseudo replicates by calculating the mean as the total 15 or did you previously calculate the mean first for each colony respectively and then calculated your initial mass? (in which case true sample size would be N=3). I would advise to keep the statistical results for different comparisons in separate columns for clearer presentation (after the values), at the moment it is very hard to clearly see what is significant.
Authors’ response: No, there was no pseudoreplication! As Table 2 shows, some groups had sample size of 14 individuals, while others had 15. These were 14 or 15 individuals per worker size-class, per species.
L225 The legend should be self-explanatory, the panels are not defined in the legend.
Authors’ response: We have defined each panel in the legend.
As far as I can tell, only Fig. 1B holds significant information with an actual relevant statistical result, justifying its graphical representation in the main manuscript. In my opinion, there is no need for the other graphs in the main manuscript or even in the supplemental material, they could just as well be summarized in the text without a full-page Figure.
Authors’ response: We have moved Figures 3 – 6 to Supplementary materials.
Moreover, in Figures 1 and 2 it is unclear to me what exactly the posthoc test contains. Are you comparing all results across a figure panel? Or just between subsets of the panel? (i.e. only between size classes, alive or dead, or across species)
Authors’ response: The comparison is just between subsets of a panel (i.e. only between worker size classes, live or dead status, and across species).
We have included the following clarification in the legends for Figures 1 & 2: “Means within each worker size-class or species with the same letter are not significantly different (P < 0.05)”.
It is also unclear in both graphs what the sample size for each bar plot was.
Authors’ response: We have included the following clarification into the Figure legend: “N = 14 or 15 individuals per worker size-class per species”.
Figure 3 Figure panels are missing. It is also unclear what the actual sample size is for each data point (with variance).
Authors’ response: We have included Figure panels for Figures 3 – 6. We have also clarified the sample size in the Figure legend by adding: “N = 14 or 15 individuals per worker size-class per species”.
Why is there only one data point for large and medium S. geminata in the fourth panel of Fig. 3? The same question also in Figure 4, 5 and 6.
Authors’ response: There were only one data point for large and medium S. geminata in the fourth panel of Fig. 3 – 6 because we plotted means.
Tables 3-8 Include sample sizes as a column.
Authors’ response: We have inserted sample sizes as a column in Tables 3 – 8.
L395-406: Please improve on readability. A sentence which is 12 lines long with multiple parentheses and side sentences is impossible to read clearly! There is also no need to keep repeating detailed results in the discussion.
Authors’ rebuttal: For the purpose of clarity to our readers, we strongly believe that the current form of the paragraph is clear and readable. Thus, we will leave the format of the paragraph as it is.
Reviewer 2 Report
In this manuscript, Ajayi et al. test different physiological conditions and tolerances of three fire ant species and a hybrid that are commonly found in the southeastern United States. Specifically, they determine total body water content (%TBW), rate of mass loss, rate of total body water loss, cuticular permeability, and desiccation sensitivity of these fire ants species. In doing so they found that %TBW was similar among species, but S. invicta and S. geminata had lower CP values, yet there were no differences in desiccation tolerance. While I appreciate the questions and amount of work that went into this project, I do have a number of comments. I have listed a few main points and additional comments in chronological order.
Overall:
- There are too many graphical elements for this manuscript (7 figures and 8 tables) making it a bit hard to read from pages 13-19. While I appreciate that the authors wish to show off their data, I think some of this information could be condensed and sent to a supplemental section to help the readability of the manuscript. Specifically tables 3-6 and figures 3-6 seemed repetitive.
- There seemed to be a lack of information on the stats from sections 3.1.1 to 3.1.4. These sections were connected to Figures 1 and 2 and while post hoc letters were created for certain tests in these figures, I did not see overall model summaries comparing different predictors and interaction terms which had been listed in the methods. I do not think a massive table is necessary for the main text but just having a p-value after a statement that something was not significant and a reference to supplemental table "X" which has the model outputs might be helpful.
- I have listed a few comments below about particular statements that caught my attention during my review that I think warrant further examination (temperate vs tropical locales, distribution based on desiccation tolerance, body size of foragers, etc.). Please take a look at those comments. One additional thing I would like to mention is fire ant behavior and how it could be important for their dominance and expansion. There is evidence that fire ants bury food resources to allow foraging during hotter parts of the day, fire ants can and will forage at night, fire ants create extensive underground tunnels, etc. There are numerous other interesting behaviors that could be incorporated into the discussion that would pair nicely with the physiological study done here. While not necessary to include them, it may help draw in additional readers that are looking to combine aspects of physiology, behavior, and ecology to better understand the distribution and potential spread of invasive species like fire ants.
Lines 15-16: Why would the hypothesis be that tropical species have lower CP (cuticular permeability) and tolerate higher levels of desiccation than temperate species? I know this is followed up with some examples in the introduction of prior studies finding these differences and this statement is repeated on line 110, but I would like to know more from the authors on why this is their hypothesis. To me, tropical species live in humid environments where desiccation shouldn’t be a huge concern.
Line 16: Are S. richteri and its hybrid actually considered temperate? Both red and black fire ants are tropical/sub-tropical. They both invaded the southeastern United States and S. richteri was actually displaced by S. invicta later. Similar question on line 64 where it may be a good place to specifically define how you are classifying temperate versus tropical (e.g greatest proportion of the population of a species found in an area, the study population came from a certain location, etc).
Line 20: LT50 isn’t defined here like the other acronyms. Definition first appears on line 88. I would just define it in the abstract as well.
Line 110: Repeat comment as above on lines 15-16. Why this particular prediction for the hypothesis? What assumptions brought you to this prediction?
Line 159: Could you expand on the explanation a little more for how confounding factors could occur from body shape and surface area?
Line 163: Is water loss calculated from the adjusted mass loss equation below on line 171 and just converted from mg to ug
Line 171: Why are different units used here (g and mg)?
Lines 217-218 and lines 220-221 are identical. Suggest rephrasing to just restate when you are talking about live versus dead ants.
Line 230: I don’t get this result “There was no significant difference in adjusted mass loss among the four species in each size (Figure 2B). Please explain as it looks like there are differences in 2B
Line 246: I don’t really understand the statement “Adjusted mass loss of small workers was greater than medium, and that of medium was greater than larger workers in all species except S. invicta x S. richteri (Figure 2E).” the letters above the bar graph in Figure 2E clearly suggest different results. Please clarify this.
Line 381: The numbers listed here do not look very similar (72 to 65). I would reword this sentence.
Line 383: At the beginning of 2nd paragraph of the discussion, the opposite is stated ( Line 378 “Large workers had significantly lower percent total body water loss than small ones in all tested species”). Please clarify.
Lines 393 to 415: While I can appreciate the sentiment here, the examples provided from other invertebrate taxa have a lot of overlap (and the authors state as much in the preceding sentences). I think stating things are more xeric or mesic though isn’t necessarily appropriate given the life history of the fire ants in this study and the habitat that these species typically occupy. If this was a comparison with S. aurea or S. amblychila then perhaps, but S. invicta is often limited by water and calling it xeric feels wrong.
Line 412: I do not believe this statement is true. While small workers do tend brood (minims), small workers with HWs around 0.7mm often make up the majority of the colony (Tschinkel 2006). These smaller workers will forage, scout, and recruit larger workers to food (Wills et al. 2018, Roeder et. al. 2020). This is the first instance of this particular statement (also listed lines 447 and 487) and I would suggest revising here and the other arguments that are based on these statements.
Line 416-423: This paragraph felt repetitive to information in the results. Is there anything here like surface area and how that scales with body size that could be useful? Maybe bringing up lines 439-443 would help.
Line 429: Unclear what “behavior of this stage in the colony ” is referring to. Please clarify.
Line 437-438: any proof of this in ants or other insects?
Line 477: S. invicta and S. geminata inhabit incredibly humid locales (gulf coast states and south eastern United States). Saying that S. richteri inhabits cooler, more humid location will need some specific examples to prove that those locations are more humid than the locations where the tropical/subtropical fire ants are found.
Lines 479-486: While abiotic conditions can limit the distribution of organisms, S. richteri did inhabit the SE united states and was pushed northward by the more aggressive S. invicta. Invicta is actually the species that is limited from moving more northward by temperature, so I am not sure desiccation in this particular case is the driving force behind the distribution of the species Northward.
Figures and Tables:
Figures 1 and 2: Just a suggestion, but I would move the letters for the panels above the graph. I was a bit confused at first.
Figure 2B has a “dead” label above it. Was this supposed to be “live”?
Figure 2E has a “live” label above it. Was this supposed to be “dead”?
Tables 3-6 or Figures 3-6: Move to supplemental material. Completely unneeded to list twice plus have stats listed on line 256.
References:
Wills et al. 2018. Correlates and Consequences of Worker Polymorphism in Ants. Annual review of Entomology 63: 575-598.
Roeder et al. 2020. The Economics of Optimal Foraging by the Red Imported Fire Ant. Environmental Entomology 49: 304-311.
Tschinkel. 2006. The Fire Ants.
Author Response
Reviewer #2
Lines 15-16: Why would the hypothesis be that tropical species have lower CP (cuticular permeability) and tolerate higher levels of desiccation than temperate species? I know this is followed up with some examples in the introduction of prior studies finding these differences and this statement is repeated on line 110, but I would like to know more from the authors on why this is their hypothesis. To me, tropical species live in humid environments where desiccation should not be a huge concern.
Authors’ response:
Some of these tropical species have become invasive species in non-tropical areas. Our paper tries to explore how this could happen.
Moreover, this is our hypothesis because:
(1) Previous studies such as Chen et al. (2014) indicated that S. invicta is more tolerant to heat and desiccation stress than S. richteri.
(2) Due to the high surface area to volume ratio of terrestrial insects, cuticular permeability and/or desiccation tolerance are important physiological traits impacting the adaptation of terrestrial species to tropical or temperate habitats.
(3) In their native land, S. invicta and S. geminata are predominantly found in tropical and subtropical habitats; while S. richteri is predominantly found in somewhat more temperate habitats (Trager 1991, Ometto et al. 2012). Furthermore, it is expected that there would be a relatively greater interspecific divergence in the patterns of selection acting on workers of these species, because adult workers most directly experience the distinct environments characterizing the different ranges occupied by S. invicta and S. geminata (namely, tropical and subtropical habitats) and S. richteri (somewhat more temperate habitats) (Trager 1991, Ometto et al. 2012).
Humid environment will provide an advantage for tropical terrestrial species to live in, compared to arid environment. Thus, desiccation may not be a huge concern in such habitat. However, when exposed to humid habitats, there is higher expectation that tropical species would adapt better than non-tropical species.
We expected that there should be relatively greater interspecific divergence in the patterns of selection acting on workers, this is because adult members of this caste (workers) most directly experience the distinct environments characterizing the different ranges occupied by the two species (namely, tropical and subtropical habitats for S. invicta and somewhat more temperate habitats for S. richteri – Trager 1991).
We know S. invicta prefers moist habitats (Allen et al. (1974) – in the article “The Red Imported Fire Ant, Solenopsis invicta; Distribution and Habitat in Mato Grosso, Brazil. Annals of the Entomological Society of America Vol. 67, # 1, pages 43- 46. However, in our study, we are trying to compare the soil-moisture preference of two sets of Solenopsis species – (S. invicta & S. geminata) to (S. invicta × S. richteri & S. richteri). We assume that the latter would prefer moist habitats more than the former group because of the current distribution in South America, the native land of S. invicta and S. richteri.
Line 16: Are S. richteri and its hybrid actually considered temperate? Both red and black fire ants are tropical/sub-tropical. They both invaded the southeastern United States and S. richteri was actually displaced by S. invicta later. Similar question on line 64 where it may be a good place to specifically define how you are classifying temperate versus tropical (e.g. greatest proportion of the population of a species found in an area, the study population came from a certain location, etc.).
Authors’ response: Rephrase S. richteri as dominant in “somewhat more temperate habitat”.
Compared to S. invicta, S. richteri is somewhat more temperate (Ometto et al. 2012, Trager 1991). However, we consider S. invicta × S. richteri in between temperate and tropical.
Furthermore, in North America, S. richteri apparently once occupied much of Alabama and northeastern Mississippi. To the south, the current North American range of S. richteri is bordered by a broad band of territory occupied by the S. invicta × S. richteri hybrid population, encompassing much of northern Alabama and Mississippi and a portion of northwestern Georgia (Trager 1991).
In North America, S. invicta occurs from the Carolinas to Florida west to Texas. Isolated populations have been found somewhat to the north of this area, and have also been found in New Mexico, Arizona and California, where they arrived with sod or nursery stock from the southeast These outliers were quickly eliminated shortly after their discovery (Trager 1991).
S. geminata is apparently native from the southeast coastal plain and Florida to Texas (probably lacking in Alabama, Mississippi and Louisiana) south through Central America to northern South America, including the coastal areas of northeastern Brazil, west through the Guianas to the Orinoco Basin, the western Amazon Basin and coastal areas of Peru. Populations of the Antilles and Galápagos (and possibly the southeastern USA) are probably introduced but have been in these areas for several centuries (Trager 1991).
Line 20: LT50 is not defined here like the other acronyms. Definition first appears on line 88. I would just define it in the abstract as well.
Authors’ response: LT50 is lethal time to kill 50% of the population. We have defined it here.
Line 110: Repeat comment as above on lines 15-16. Why this particular prediction for the hypothesis? What assumptions brought you to this prediction?
Authors’ response: The assumptions are that:
(1) Previous studies such as Chen et al. (2014) indicated that S. invicta is more tolerant to heat and desiccation stress than S. richteri.
(2) Due to the high surface area to volume ratio of terrestrial insects, cuticular permeability and/or desiccation tolerance are important physiological traits impacting the adaptation of terrestrial species to tropical or temperate habitats.
(3) In their native land, S. invicta and S. geminata are predominantly found in tropical and subtropical habitats; while S. richteri is predominantly found in somewhat more temperate habitats (Trager 1991, Ometto et al. 2012). Furthermore, it is expected that there would be a relatively greater interspecific divergence in the patterns of selection acting on workers of these species, because adult workers most directly experience the distinct environments characterizing the different ranges occupied by S. invicta and S. geminata (namely, tropical and subtropical habitats) and S. richteri (somewhat more temperate habitats) (Trager 1991, Ometto et al. 2012).
Humid environment will provide an advantage for tropical terrestrial species to live in, compared to arid environment. Thus, desiccation may not be a huge concern in such habitat. However, when exposed to humid habitats, there is higher expectation that tropical species would adapt better than non-tropical species.
Line 159: Could you expand on the explanation a little more for how confounding factors could occur from body shape and surface area?
Authors’ response: Insects shrivel when they dry up and the sclerites start to overlap. Thus, the surface area of the insect changes during the drying process. We use the difference between 0- and 2-hour masses to calculate CP since there is minimal change in surface area between these two times.
Line 163: Is water loss calculated from the adjusted mass loss equation below on line 171 and just converted from mg to ug
Authors’ response: Water lost in the numerator of the CP equation is mass loss between T0 and T2 (just like in line 171). In line 163, we kept the ‘units’ in g. By the time SA, time, and saturation deficit are accounted for, the area specific mass loss is very small. Therefore, we use units of ug. It is just a lot easier to talk about a CP of 2 versus a CP of 0.000002.
Line 171: Why are different units used here (g and mg)?
Authors’ response: g is for body mass, while mg is for water lost.
Lines 217-218 and lines 220-221 are identical. Suggest rephrasing to just restate when you are talking about live versus dead ants.
Authors’ response: Lines 217 – 218 have been rephrased to: “There was no significant difference in %TBW between tropical/subtropical and temperate species (Figure 1C) when worker sizes were combined”.
Line 230: I don’t get this result “There was no significant difference in adjusted mass loss among the four species in each size (Figure 2B). Please explain as it looks like there are differences in 2B
Authors’ response: The section has been rephrased to:
“Temperate S. richteri had significantly greater adjusted mass loss than the tropical S. invicta (Figure 2A). There was no significant difference in adjusted mass loss among the four species in each size (Figure 2C). Small workers of S. invicta × S. richteri and S. geminata had significantly greater mass loss than large workers (Figure 2E). Across all species, the ranking of adjusted mass loss was small > medium > large (Figure 2E).”
Line 246: I don’t really understand the statement “Adjusted mass loss of small workers was greater than medium, and that of medium was greater than larger workers in all species except S. invicta × S. richteri (Figure 2E).” the letters above the bar graph in Figure 2E clearly suggest different results. Please clarify this.
Authors’ response: This section has been rephrased to: “There were significant differences in adjusted mass loss between temperate (S. richteri and S. invicta × S. richteri) and sub-tropical (S. geminata) ant workers both in medium and large sizes (Figure 2B). Temperate S. invicta × S. richteri had significantly greater adjusted mass loss than tropical species S. invicta and S. geminata for both medium and large workers (Figure 2B). Ranking of species by adjusted mass loss of medium and large sizes was S. invicta × S. richteri > S. richteri > S. invicta > S. geminata (Figure 2B). There were significant differences in adjusted mass loss among the sizes within each species (Figure 2E). Adjusted mass loss of small workers was greater than medium, and that of medium was greater than large workers in all species except S. invicta × S. richteri (Figure 2E). Ranking of sizes by adjusted mass loss was small > medium > large for all the four Solenopsis species (Figure 2E).”
Line 381: The numbers listed here do not look very similar (72 to 65). I would reword this sentence.
Authors’ response: This section has been reworded to: “… was close to that in …”
Line 383: At the beginning of 2nd paragraph of the discussion, the opposite is stated (Line 378 “Large workers had significantly lower percent total body water loss than small ones in all tested species”). Please clarify.
Authors’ response: Thanks. It was an error. The “greater” or “higher” have been changed to “lower”.
Lines 393 to 415: While I can appreciate the sentiment here, the examples provided from other invertebrate taxa have a lot of overlap (and the authors state as much in the preceding sentences). I think stating things are more xeric or mesic though is not necessarily appropriate given the life history of the fire ants in this study and the habitat that these species typically occupy. If this was a comparison with S. aurea or S. amblychila then perhaps, but S. invicta is often limited by water and calling it xeric feels wrong.
Authors’ response: We have rephrased this section and deleted the “xeric” and “mesic” comparisons.
Line 412: I do not believe this statement is true. While small workers do tend brood (minims), small workers with HWs around 0.7mm often make up the majority of the colony (Tschinkel 2006). These smaller workers will forage, scout, and recruit larger workers to food (Wills et al. 2018, Roeder et. al. 2020). This is the first instance of this particular statement (also listed lines 447 and 487) and I would suggest revising here and the other arguments that are based on these statements.
Authors’ response: Lines 412, 447, and 487, and the arguments based on the statements have all been revised to reflect the reviewer’s suggestion that all worker size-classes engage in foraging.
Line 416-423: This paragraph felt repetitive to information in the results. Is there anything here like surface area and how that scales with body size that could be useful? Maybe bringing up lines 439-443 would help.
Authors’ response: Lines 439 – 443 have been brought up as suggested by the reviewer.
Line 429: Unclear what “behavior of this stage in the colony” is referring to. Please clarify.
Authors’ response: It is foraging behavior. The statement has been revised.
Line 437-438: any proof of this in ants or other insects?
Authors’ response: Actually, there is no ‘proof’, just speculations. Therefore, we have provided the following 2 references: Winston (1967) & Winston and Beament (1969) in these Lines. These 2 references infer that the presence of a ‘cuticular water pump’ controlled by epidermal cells removes water from the cuticle, which actively reduces water loss (Winston 1967, Winston and Beament 1969).
Furthermore, Edney (2012) stated: “In Porcellio, (wood-lice) according to Salminen and Lindqvist (1972) and Lindqvist et al. (1972), the water content of the cuticle is maintained at a constant level during transpiration, either by pumping from within or, additionally, by a spreading over the cuticle of water regurgitated from the mouth (Lindqvist, 1971, 1972).”
Line 477: S. invicta and S. geminata inhabit incredibly humid locales (gulf coast states and south eastern United States). Saying that S. richteri inhabits cooler, more humid location will need some specific examples to prove that those locations are more humid than the locations where the tropical/subtropical fire ants are found.
Authors’ response: We have rephrased S. richteri as dominant in somewhat more temperate habitats. Furthermore, compared to S. invicta, S. richteri is somewhat more temperate (Ometto et al. 2012, Trager 1991). However, we consider S. invicta × S. richteri in between temperate and tropical.
Ometto et al. (2012) inferred that there should be relatively greater interspecific divergence in the patterns of selection acting on Solenopsis fire ant workers because workers most directly experience the distinct environments characterizing the different ranges occupied by the two species (namely, tropical and subtropical habitats for S. invicta and somewhat more temperate habitats for S. richteri (Trager 1991).
In North America, S. invicta occurs from the Carolinas to Florida west to Texas. Isolated populations have been found somewhat to the north of this area, and have also been found in New Mexico, Arizona and California, where they arrived with sod or nursery stock from the southeast These outliers were quickly eliminated shortly after their discovery (Trager 1991).
S. geminata is apparently native from the southeast coastal plain and Florida to Texas (probably lacking in Alabama, Mississippi and Louisiana) south through Central America to northern South America, including the coastal areas of northeastern Brazil, west through the Guianas to the Orinoco Basin, the western Amazon Basin and coastal areas of Peru. Populations of the Antilles and Galápagos (and possibly the southeastern USA) are probably introduced but have been in these areas for several centuries (Trager 1991).
Lines 479-486: While abiotic conditions can limit the distribution of organisms, S. richteri did inhabit the SE united states and was pushed northward by the more aggressive S. invicta. S. invicta is actually the species that is limited from moving more northward by temperature, so I am not sure desiccation in this particular case is the driving force behind the distribution of the species Northward.
Authors’ response: We have rephrased the statement to reflect the reviewer’s suggestion that desiccation tolerance could be part of a complex of physiological and behavioral factors behind the distribution of the species.
Figures and Tables:
Figures 1 and 2: Just a suggestion, but I would move the letters for the panels above the graph. I was a bit confused at first.
Authors’ response: We have moved the letters for the panels above the graph in Figures 1 & 2.
Figure 2B has a “dead” label above it. Was this supposed to be “live”?
Authors’ response: It was wrongly labelled but has been correctly labelled.
Figure 2E has a “live” label above it. Was this supposed to be “dead”?
Authors’ response: It was wrongly labelled but has been correctly labelled.
Tables 3-6 or Figures 3-6: Move to supplemental material. Completely unneeded to list twice plus have stats listed on line 256.
Authors’ response: Figures 3 – 6 have been moved to supplementary material.
Reviewer 3 Report
The authors examine cuticular permeability, dessication tolerance, total body water content, and water loss rate in two species of fire ants in tropical and temperate ranges. Overall, the paper summarizes previous work well and present novel data that expands our understanding of the species.
In a few places, the text needs some polishing (e.g., lines 140-141 are in the future tense), but overall is readable and clear.
I have no issues with the science, methods, or data analysis. The results follow logically from the methodology. The figures are clear and easy to interpret. The discussion contextualizes the results well.
Author Response
The authors examine cuticular permeability, dessication tolerance, total body water content, and water loss rate in two species of fire ants in tropical and temperate ranges. Overall, the paper summarizes previous work well and present novel data that expands our understanding of the species.
In a few places, the text needs some polishing (e.g., lines 140-141 are in the future tense), but overall is readable and clear.
Authors’ response: We have polished this statement and rephrased it from past tense to present tense.
I have no issues with the science, methods, or data analysis. The results follow logically from the methodology. The figures are clear and easy to interpret. The discussion contextualizes the results well.